# Untargeted metabolomics analysis reveals the metabolic disturbances and exacerbation of oxidative stress in recurrent spontaneous abortion

AiNing Wu[1☯], YanHui Zhao[2☯], RongXin Yu[1], JianXing Zhou[3], Ya Tuo[3]*

1 Obstetrics and Gynecology, The Affiliated Hospital of Inner Mongolia Medical University, Huhhot, China, 2 Obstetrics department, Chifeng Municipal Hospital, Chifeng, China, 3 Department of Reproductive Medicine Centre, The Affiliated Hospital of Inner Mongolia Medical University, Huhhot, China

☯ These authors contributed equally to this work.
* nmgty81@163.com

## Abstract

### Background

Recurrent spontaneous abortion (RSA) is characterized by the occurrence of two or more consecutive spontaneous abortions, with a rising prevalence among pregnant women and significant implications for their physical and mental well-being. The multifaceted etiology of RSA has posed challenges in unraveling the molecular mechanisms underlying that underlie its pathogenesis. Oxidative stress and immune response have been identified as pivotal factors in the development of its condition.

### Methods

Eleven serum samples from healthy pregnant women and 17 from RSA were subjected to liquid chromatography/mass spectrometry (LC-MS) analysis. Multivariate statistical analysis was employed to excavate system-level characterization of the serum metabolome. The measurement of seven oxidative stress products, namely superoxide dismutase (SOD), catalase (CAT), malonaldehyde (MDA), glutathione (GPx), glutathione peroxidase (GSH), oxidized glutathione (GSSG), heme oxygenase (HO-1), was carried out using ELISA.

### Results

Through the monitoring of metabolic and lipid alternations during RSA events, we have identified 816 biomarkers that were implicated in various metabolic pathways, including glutathione metabolism, phosphonate and phosphinate metabolism, nucleotide metabolism, sphingolipid metabolism, lysine degradation and purine metabolism, etc. These pathways have been found to be closely associated with the progression of the disease. Our finding indicated that the levels of MDA and HO-1 were elevated in the RSA group compared to the control group, whereas SOD, CAT and GPx exhibited a contrary pattern. However, no slight difference was observed in GSH and GSSG levels between the RSA group and the control group.

**Data Availability Statement:** All relevant data are within the paper and its Supporting Information files.

**Funding:** This study was funded by grants from Technology Engineering Million Project of the Affiliated Hospital of Inner Mongolia Medical University (No.YKD2020KJBW(LH)037), Program for Young Talents of Science and Technology in Universities of Inner Mongolia Autonomous Region (No. NJYT22006), Inner Mongolia Natural Science foundation (No.2021MS08022), and Doctoral Fund Project of Affiliated Hospital of Inner Mongolia Medical University (No. NYFY BS202137). The funders had no role in study design, data collection and analysis, decision to publish, or preparation of the manuscript.

**Competing interests:** The authors have declared that no competing interests exist.

## Conclusion

The manifestation of RSA elicited discernible temporal alternations in the serum metabolome and biochemical markers linked to the metabolic pathways of oxidative stress and immune response. Our investigation furnished a more comprehensive analytical framework encompassing metabolites and enzymes associated with oxidative stress. This inquiry furnished a more nuanced comprehension of the pathogenesis of RSA and established the ground work for prognostication and prophylaxis.

## Introduction

RSA, which is characterized by the incidence of two or more spontaneous abortion, is a prevalent pregnancy complication affecting 1–2% of all pregnant women [1]. Various factors have been implicated in the advancement of RSA, such as chromosome abnormality, maternal reproductive tract abnormality, maternal endocrine abnormality, immune function abnormality, and unhealthy lifestyle practices [2]. However, approximately 50% of RSA cases remain unexplained, indicating that this disorder is multifactorial and that there are numerous inconsistencies and uncertain ties in its diagnosis and treatment in clinical settings. Consequently, there is a processing need to unravel the pathogenesis underlying RSA and gain a comprehensive understanding of its etiology.

Metabolomics is a scientific discipline that investigates the dynamic metabolic alterations of an organism or a tissue cell system through the analysis of group indices [3]. This field concentrates on metabolites that have been perturbed by external stimuli, facilitating the establishment of connections with phenotypes [4]. The occurrence of RSA is an anonymous event that exerts significant pressure on affected families. However, the precurse or information regarding the modified metabolic profile between RSA patients and healthy controls can be utilized for the diagnosis and treatment of disease. Metabolomics has been utilized to identify distinct metabolites that can serve as novel biomarkers for predicting RSA [5]. Both non-volatile targeted metabolites and volatile non-targeted metabolites have been employed on a mass spectrometer (MS) platform to differentiate the RSA and healthy groups [6]. The liquid chromatogram (LC) platform's extensive coverage and board dynamic range of metabolites are its primary advantages, making it the preferred method for capturing the disrupted discipline of the organism or cell. This study utilized a LC-MS based metabolomics approach to examine alterations in metabolic profiles between pregnant women with controls. The primary objective of this investigation was to elucidate the underlying mechanisms contributing to the onset and progression of RSA by identifying potential metabolites as an initial step. Additionally, the findings of this study may serve as a basis for the diagnosis and treatment of pregnant women.

## Methods

### Ethics statement

The protocol of this study was approved by the ethics committee of the Affiliated Hospital of Inner Mongolia Medical University (series number: s.2022055) and all eligible individuals provided written informed consent. All information about the participants was anonymized.

## Study subjects

Seventeen individuals diagnosed with RSA and 11 healthy pregnant women as a control group were recruited from the Affiliated Hospital of Inner Mongolia Medical University between November 2022 and March 2023. The RSA group consisted of patients who had experienced more than two unexplained and consecutive spontaneous abortions at less than 12 weeks of gestation without any additional symptoms. Those who had genital abnormalities; chronic hypertension; diabetes; liver, kidney, cardiovascular, and thyroid disease; autoimmune disease; or infectious were excluded. The control group was comprised of pregnant woman in the first trimester undergoing a health checkup at a gestational age that correspond to that of the RSA group. The control group patients were administered no medication and did not undergo any assisted reproductive technology.

## Sample collection and preparation

Ethics standards of the Affiliated Hospital of Inner Mongolia Medical University were followed during the collection of serum samples. Five milliliters of venous blood were obtained in the morning, left to coagulate at room temperature for 20 minutes, and then centrifuged to eliminate the clot. The processed samples were divided into aliquots and preserved in a -80˚C refrigerator for future utilization. The serum samples underwent a thawing process followed by vortexing for a duration of 10 seconds. To conduct metabolomics analysis, a 50 μL serum sample was extracted and subjected to deproteination using a 300 μL acetonitrile-methanol internal standard solution (1:4, v:v). The resulting mixture was vortexed for 3 minutes and subsequently centrifuged at 12 000 rpm for 10 minutes at 4˚C. The processed supernatant was then analyzed using LC/MS. To assess the reliability and consistency of the experimental procedure, quality control (QC) samples were prepared by obtaining equal samples. To monitor the repeatability of analysis process, a single QC sample was incorporated into every 14 tests analyses during the analysis phase.

## Metabolomic analysis using liquid chromatography-mass spectrometry (LC-MS)

The High-performance liquid chromatography time-of-flight mass spectrometry (HPLC-TOF-MS) analysis was conducted utilizing an LC-30A HPLC system (Shimadzu, Kyoto, Japan) coupled with a TripleTOF 6600+ spectrometer (AB Sciex, Framingham, MA, USA) equipped with an electrospray ionization source (ESI). To achieve a comprehensive coverage of metabolites, the UPLC BEH C18 (1.8μm, 2.1mm* 100mm) was employed for sample separation. The temperature of the column oven was set to 40˚C, while the flow rate and injection volume was set at 0.40 mL/min and 2 μL, respectively. The optimized mobile phase comprised 0.1% formic acid in pure water (A) and 0.1% formic acid in acetonitrile (B), and both positive and negative ionization modes were utilized for the analyses across the mass range of 50 to 1200 m/z. The detailed parameters of the gradient elution and the parameters were listed in S1 Table. The stability of the analysis and detection platforms were evaluated by QC samples. The QC group demonstrated favorable repeatability and consistency in chromatographic separation, as evidenced by relative standard deviations of the peel areas below 10% (S2 Table).

## Data pretreatment and statistical analysis

The original LC-MS data file underwent conversion to mzML format through the utilization of Proteo Wizard software. Subsequently, XCMS program, a R-based platform for LC-MS data processing and visualization, was employed to process the mzML output files, including peak

extraction, peak alignment, and retention time correction. MS peak list alignment was conducted with mass tolerance (RT) tolerance values set at 0.25 Da and 30 s, respectively. The objective of this procedure was to mitigate RT shifts and prevent superfluous signals. The algorithm and parameters employed for peak detection included a minimum peak width of 5s, maximum peak width of 20 s, ppm deviation of 5 ppm, and a signal-to-noise threshold of 4. Peaks with a detection rate below rate below 50% in each sample group were eliminated, and the refined data was subsequently subjected to XCMS with various integrated statistical analysis modules. Principal component analysis (PCA) was carried out using the prcomp function within R (www.r-project.org). Prior to unsupervised PCA, the data underwent unit variance scaling. The PCA module was utilized as an impartial statistical technique to examine distribution and visualize the interrelationships among samples. In order to identify potential metabolites that distinguish RSA patients from the control group, orthogonal partial least squares data analysis (OPLS-DA) was implemented. The selection on of differential metabolites was based on algorithms utilizing variable importance in projection (VIP) values, with variables possessing a VIP value greater than 1 deemed to contribute significantly to sample separation. To confirm the statistical significance of the differential metabolites, a student's t-test was conducted between groups, with a p-value less than 0.05. Furthermore, the fold change (FC) was utilized as a valuable indicator to monitor the change trend between the RSA group and control group.

## Metabolites identification

Metabolic identification data was procured through a comprehensive search of the laboratory's proprietary database, as well as public available database including HMDB (https://hmdb.ca), AI database-local libraries based on machine learning models [7], and metDNA (metdna.zhulab.cn).

## Pathway analysis and interaction networks

The identified metabolites were subsequently annotated utilizing the KEGG compound database (https://www.genome.jp/kegg/compound/), and the annotated metabolites were mapped to the KEGG Pathway database (https://www.genome.jp/kegg/pathway.html). The identification of significantly enriched pathways was accomplished by means of a hypergeometric test's p-value for a given list of metabolites.

## Levels of superoxide dismutase, catalase, malonaldehyde, glutathione, glutathione peroxidase, oxidized glutathione, hemeoxygenase-1

The enzymatic activities of glutathione (GSH), glutathione peroxidase (GPx), and oxidized glutathione (GSSG) were determined in serum samples based on the glutathione metabolism pathway. This was accomplished using a spectrophotometer and the Wendel modified method [8]. Additionally, the enzymatic activity of CAT activity was determined in serum samples using a spectrophotometer and the Aebi method [9], while the enzymatic activity of SOD was determined using a spectrophotometer and the Misa and Fridovich protocols [10]. Serum MDA levels were assessed through spectrometry utilizing Thiobarbituric acid reactive substances (TBARS) [11]. The quantification of hemeoxygenase-1 (HO-1) was performed via commercially available enzyme-linked immunosorbent assay (ELISA) kits in accordance with the manufacturers guide.

## ELISA assay

The concentration of IL-2 was determined by Elisa according to the method of Li's method [12].

## Statistical analysis

The statistical analysis was performed utilizing the R studio program (version 4.2.0), with the student's t-test test employed to calculate group differences. A significance level of $p \leq 0.05$ was utilized to assess differences between groups.

# Results

## Study participants

The clinical information of the enrolled participants detailed is presented in **S3 Table**, including the average maternal age, gestational age, body weight, and body mass index (BMI) of patients in the RSA group, which were 33.30±4.60 years, 7.70±1.33 weeks, 65.71±6.29 kg and 24.65±2.16 kg/m$^2$, respectively (**S1 Fig**). The control group exhibited an average maternal age of 30.72±2.41 years, gestational age of 12.27±0.54 weeks, and body weight of 57.54±6.62kg. Statistically significant differences were observed in gestational age and body weight, with p-values less than 0.05. No significant difference was shown in age (p = 0.10). The number of abortions mentioned in this study were more than twice. Furthermore, a comprehensive analysis of the clinical manifestations of patients with RSA was conducted to investigate the impact of these adverse factors on disease occurrence. In omics research, intra-group differences of samples can interfere with the recognition of inter-group signals, while too small a sample size can amplify intra-group differences. The limited sample size precluded the statistical analysis of adverse factors to determine their association with disease.

## Serum metabolomics and reproducibility of the analysis

The representative serum metabolomic skeleton map both in positive and negative modes were displayed in **Fig 1**. The application of the PCA model facilitated the visualization of sample groupings based on the filtered and normalized datasets obtained through LC-MS analysis. The reliability of the experimental procedures and the stability and reproducibility of the analysis were evaluated using the distribution of QC samples, as depicted in **Fig 2**. The complete coincidence of two QC samples serves as evidences of the robustness of the entire experimental procedure and analysis process.

## Global metabolomic changes with RSA

To achieve a comprehensive coverage of metabolites during RSA occurrence, a common reverse C18 column was employed for the separation of the biological sample. The C18 column was selective for metabolites with low or non-polar polarity. The feasibility of modeling and identification of metabolites contributing to sample modeling was assessed using PCA and OPLS-DA modes. The results of PCA indicated a distinct metabolome profile between the NC group and the RSA group, with good separation between the two groups. The positive and negative modes of the PCA model yielded PC1 and PC2 parameters of 24.09% and 13.01%, and 20.04% and 9.09% respectively, indicting a satisfactory level of model explanation. The rigorous supervised analysis method, Orthogonal Partial Least Squares Data Analysis (OPLS-DA), demonstrated a distinct separation between the NC group and RSA group.

To ensure the reliability of the model and prevent overfitting, external verification through permutation testing was conducted. The model's performance was deemed perfect, as evidenced by the R2Y and Q2 parameters of 0.971 and 0.81, 0.995 and 0.881 in positive and negative mode, respectively, as depicted in **Fig 3**. In clinical research, a threshold exceeding 0.4 is considered an acceptable range.

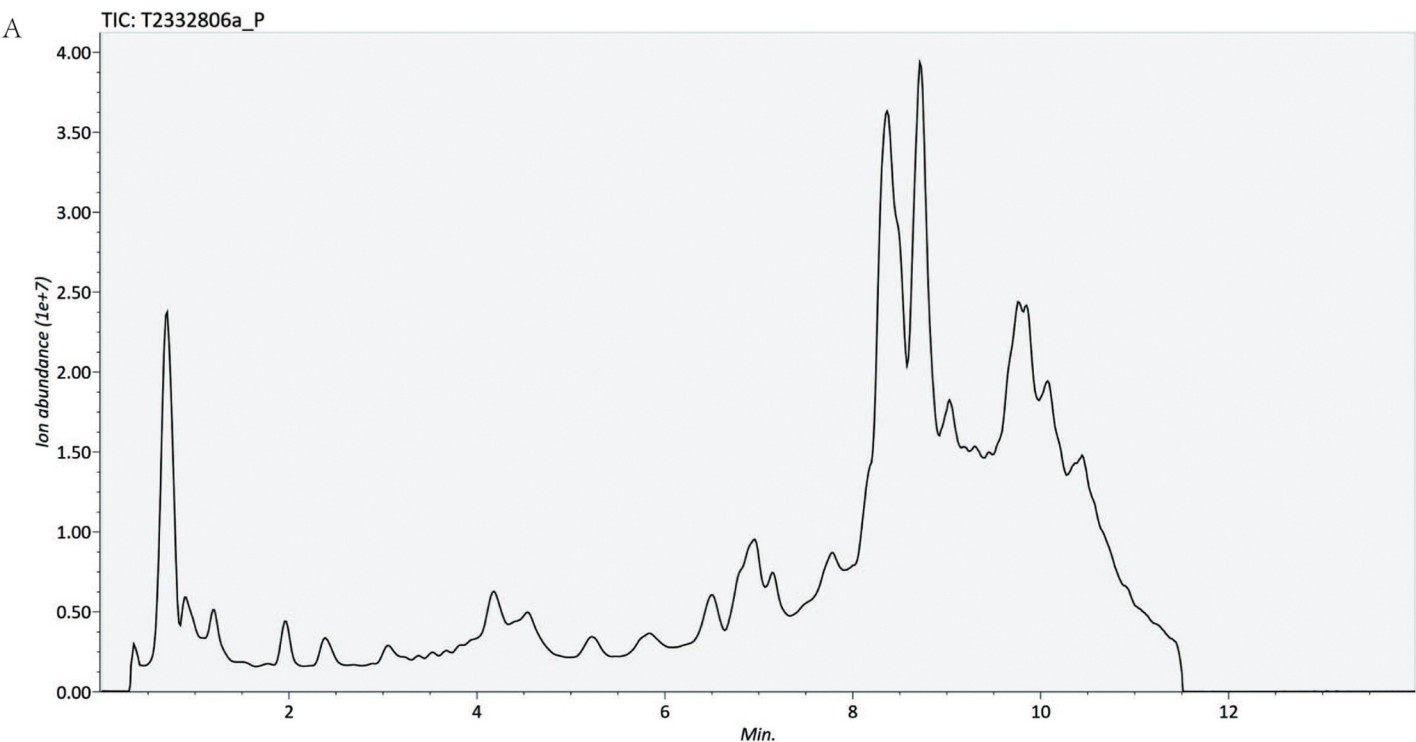

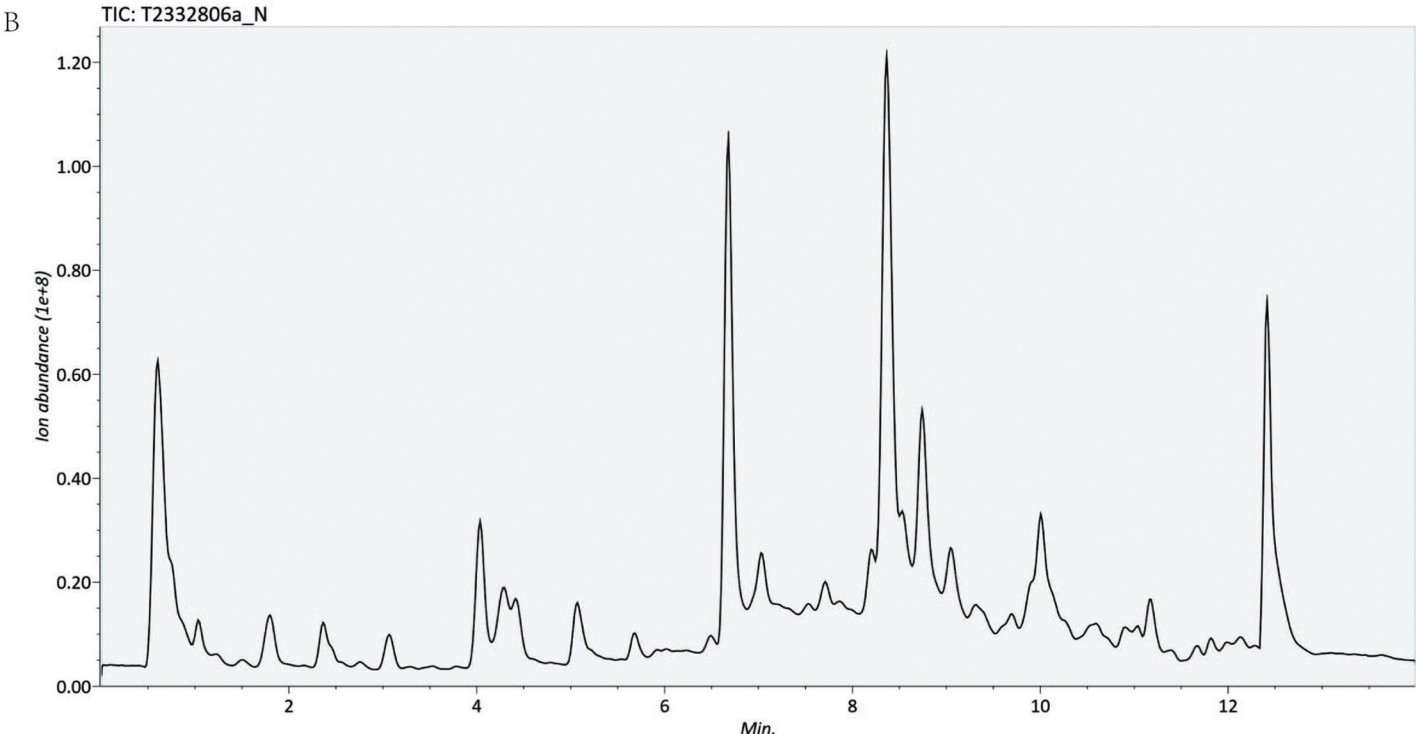

**Fig 1.** (A) The positive and (B) negative total ion current (TIC) chromatograms of the serum sample from an RSA patient.

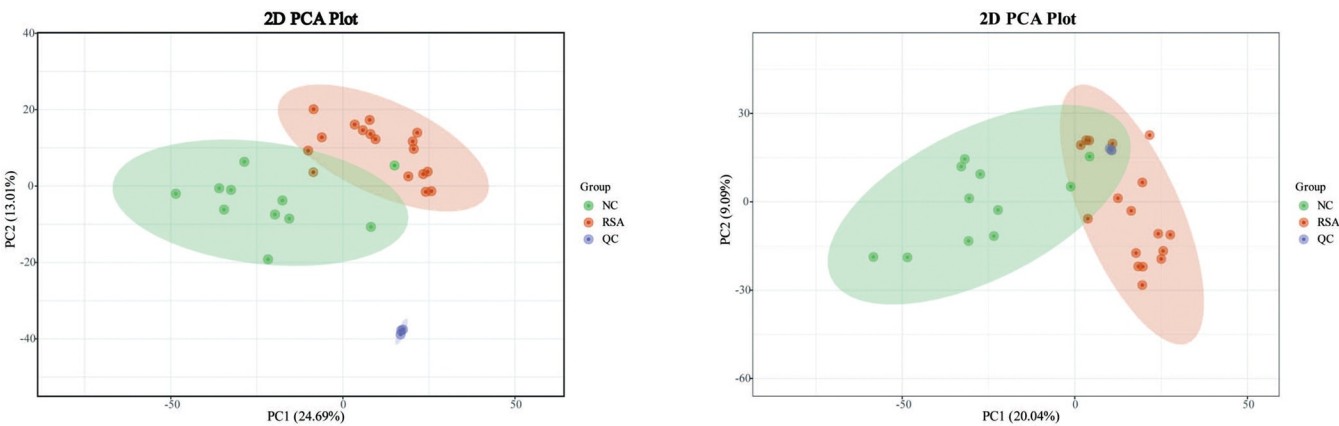

**Fig 2.** Score plots (left) in positive ion mode and (right) negative mode classifying RSA, control and QC group. (positive, PC1 = 24.69%, PC2 = 13.01%; negative, PC1 = 20.04%, PC2 = 9.09%).

## Identification of significantly altered metabolites and pathways

Approximately 1739 and 2724 metabolites were identified via LC-MS using C18 column in positive and negative mode respectively (**S4 Table**). Of these metabolites, 1189 and 1384 were confidently identified through the determination of secondary fragment structures, thereby ensuring the accuracy of metabolite identification. Additionally, 549 and 267 metabolites were structurally identified via LC-MS/MS using discriminant analysis, with a VIP value >1 and a t-test p-value < 0.05 serving as the criteria. The identified metabolites encompassed a diverse range of classes, including amino acids (AA), phosphatidycholine (PC), oxidized lipids, sphingonsine, carbohydrate, hormone, ketones, fatty acids (FA), sphingomyelin (SM), nucleotide and their metabolites, flavonoids, phosphatidyl ethanolamine (PE), acylcarnitine, ceramide, choline, vitamin. Notably, approximately 200 of these metabolites were drugs or metabolites of drugs, likely stemming from prior interventions for pregnant women with RSA. As our investigation did not account for the influence of pharmaceuticals, merely focused on the relationship between oxidative stress, immune response and RSA, these drug metabolites were excluded from our analysis. The classification information of metabolites was shown in **Fig 4**. Detailed information was listed in **S4 Table**. These metabolites exhibited diverse trends, with those from the RSA group exhibiting greater changes, compared to the control group. This suggest that they may serve as potential biomarkers and contribute to the understanding of disease pathogenesis.

The integration of potential metabolites was performed using KEGG pathway enrichment in R studio program. The results indicated that the biosynthesis of nucleotide sugars, amino sugar and nucleotide sugar metabolism, glutathione metabolism, butanoate metabolism, phosphonate and phosphinate metabolism, purine metabolism, glycolysis, sphingolipid metabolism and lysine degradation were dominant disordered metabolic pathway. The metabolic network analysis revealed that the most significant alternations were observed in the lipid metabolism and glutathione metabolism pathways, which are known to be linked with oxidative stress. A comprehensive representation of the metabolic pathways was presented in **Fig 5**.

## Oxidative stress function testing

Given the crucial role of the phospholipid metabolism and glutathione metabolism pathways in oxidative stress, the levels of seven oxidative stress markers (MDA, SOD, CAT, HO-1, GPx, GSH, and GSSG) were quantified using ELISA in RSA samples to corroborate the relationship

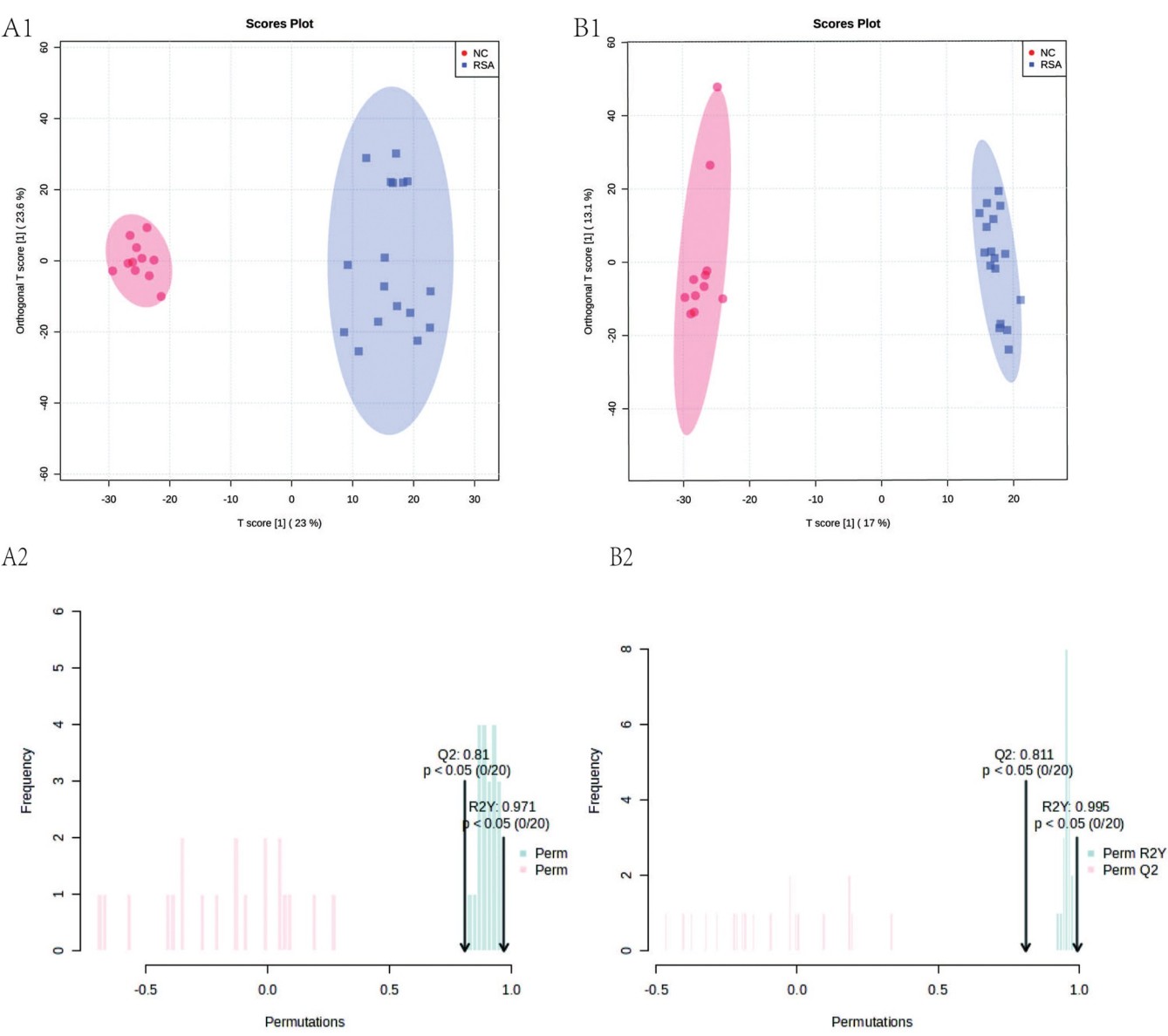

**Fig 3. OPLS-DA score plots of RSA group and control group.**

between oxidative stress and RSA severity. The levels of MDA and HO-1 in the serum samples from the patients with RSA were significantly higher compared to the control group. Conversely, SOD, CAT and GPx exhibited an elevated expression in control samples. The alterations in the aforementioned indices, which are associated with oxidative stress, were graphically represented in **Fig 6**. The GSH and GSSG levels showed no differences between the RSA and control groups. A systematic pathway that composed metabolites involved in oxidative stress as well as immune response and the related regulatory enzyme series was shown in **Fig 7**.

## Discussion

RSA, a prevalent complication of pregnancy with multifaceted etiology, is characterized by abortion symptoms before 12 weeks of gestation in most patients [13]. Presently, there is no

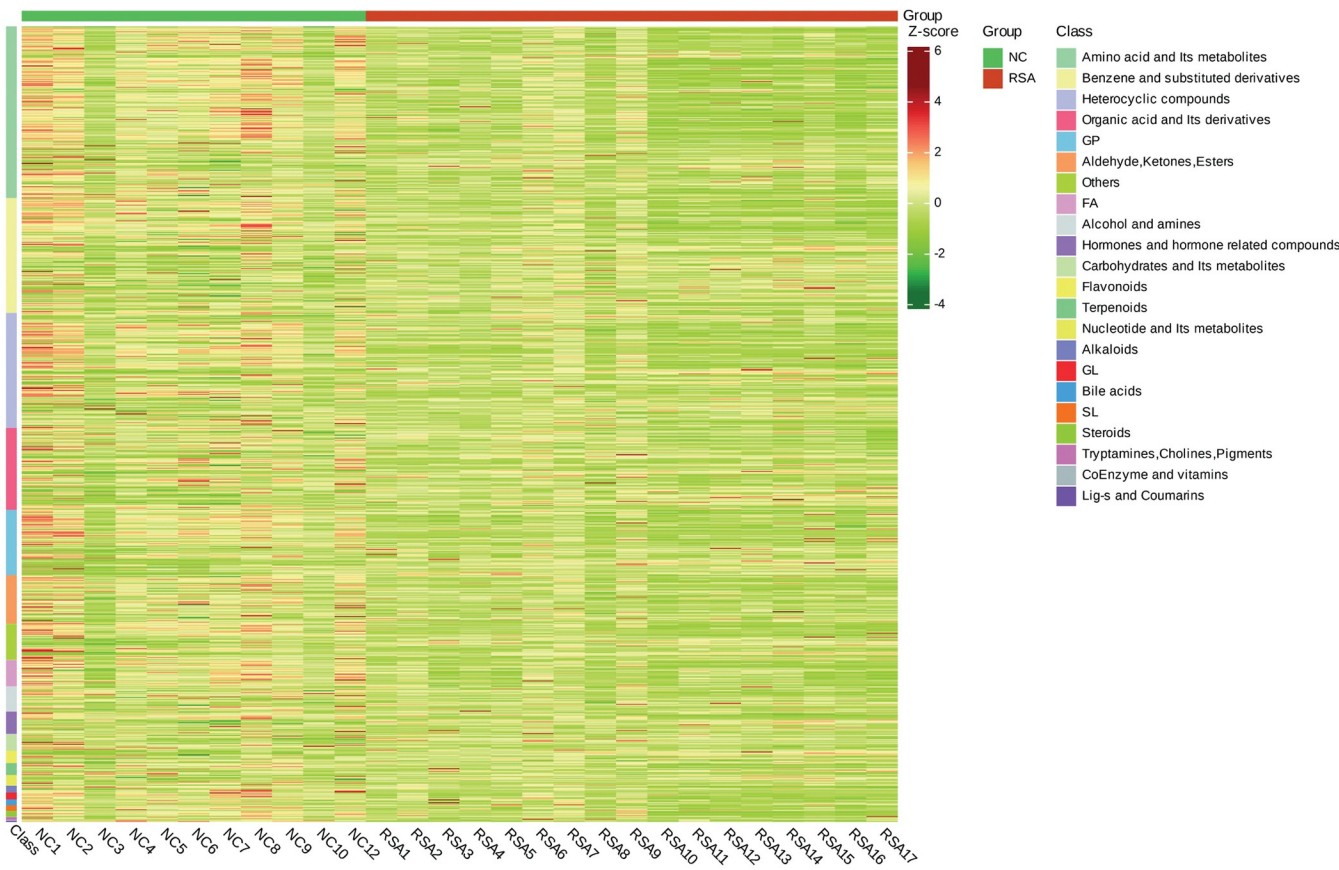

**Fig 4. Clustering analysis of the differential metabolites.**

targeted therapy available and the rate of pregnancy failure is high, causing significant physical and psychological stress to patients [14]. The etiology of RSA in women is intricate, with factors such as genital abnormalities, gene abnormalities, and the host 's immune mechanism playing crucial roles. The investigation of the molecular mechanism underlying RSA is of paramount importance in enhancing our comprehensive of the metabolic disruptions experienced by pregnant women. In this study, metabolomics was employed to elucidate the comprehensive metabolomic alterations in RSA patients, leading to the identification of modified metabolic pathways during RSA onset. Subsequently, we conducted validation experiments on select regulatory enzymes in the central pathways. The sample size of the cohort (17 RSA women and 11 healthy pregnant women) was conducted, dynamic changes were captured, resulting in the extraction of dependable differential information. Through the application of multivariate statistical analysis, a distinct separation between the RSA group and control group was observed. The potential biomarkers were shown in both positive and negative modes and contributed to the separation of RSA group and control group. Our investigation revealed a significant alteration in approximately 800 serum metabolites associated with human metabolism, including immune response and oxidative stress reaction. The identified metabolites encompassed various pro-inflammatory and oxidative stress mediators, indicating an immune response storm initiated by the host in response to abortion. While the identification of these diagnostic metabolites from a limited sample size is promising, extensive experimentation is required to assess the reliability of the findings.

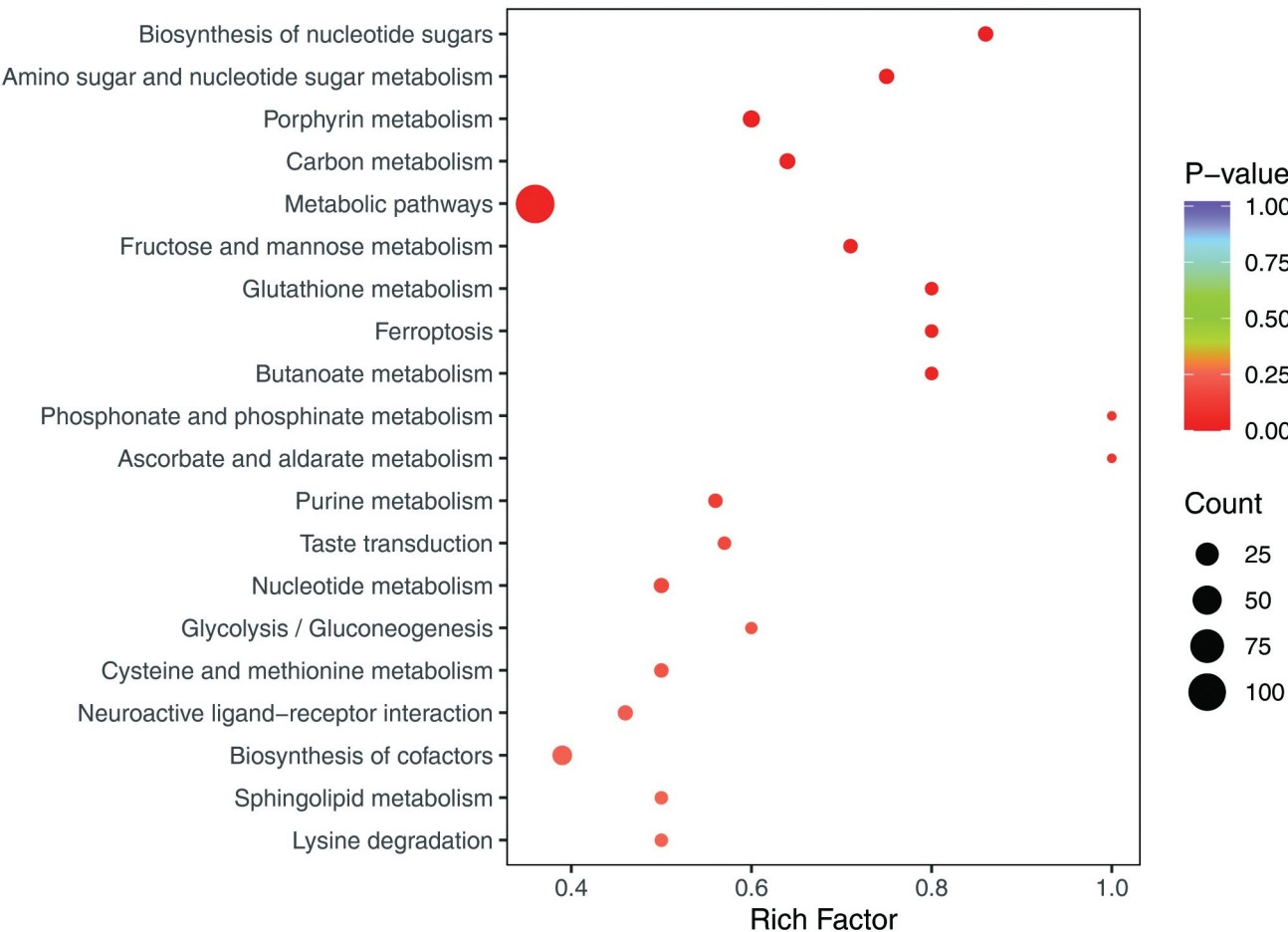

**Fig 5. KEGG enrichment pathway based on the significant metabolites detected in RSA and control group.**

RSA is typically accompanied by metabolic alterations that entail the progression of pro-inflammatory mechanisms and oxidative stress [15]. Oxidative stress is a widely recognized factor that impacts lipid metabolism [16, 17]. The presence of oxidative stress triggers the activation of enzyme systems, including phospholipase and lipoxygenases (LOX), which play a role in lipid and derivative metabolism, leading to lipid metabolism disorders. Our study found varying degrees of glyceryl phosphatide (GP), glycerolipid (GL), and sphingolipid (SL) content. Additionally, research has demonstrated that lipid mediators are not only a consequence of oxidative stress but also an inducement for modulating processes [18]. Sphingomyelin serves as a crucial constituent of membrane phospholipids and a reservoir of ceramides [19]. Our findings demonstrated an upward trend in the levels of 1-beta-D-Galactosylsphingosine and sphinganine 1-phosphate, indicating the activation of oxidative stress. Ceramide plays a significant role in physiological processes and has the potential to induce oxidative stress [20]. The disruption in the levels of N-Acetyl-Neuraminic Acid, N-octanoylsphingosine 1-phosphate, and Galabiosylceramide (d18:1/9Z-18:1) as intermediates of ceramide further suggests the occurrence of oxidative stress. The process of lipid oxidation results in the production of a range of downstream metabolites. The present investigation identified 11 metabolites associated with the oxidation of lipids, including eicosa-8,11,14-trien-5-ynoic acid, 12-Keto-leukotriene B4, and leukotriene D4, which are all metabolites of arachidonic acid (AA). AA is

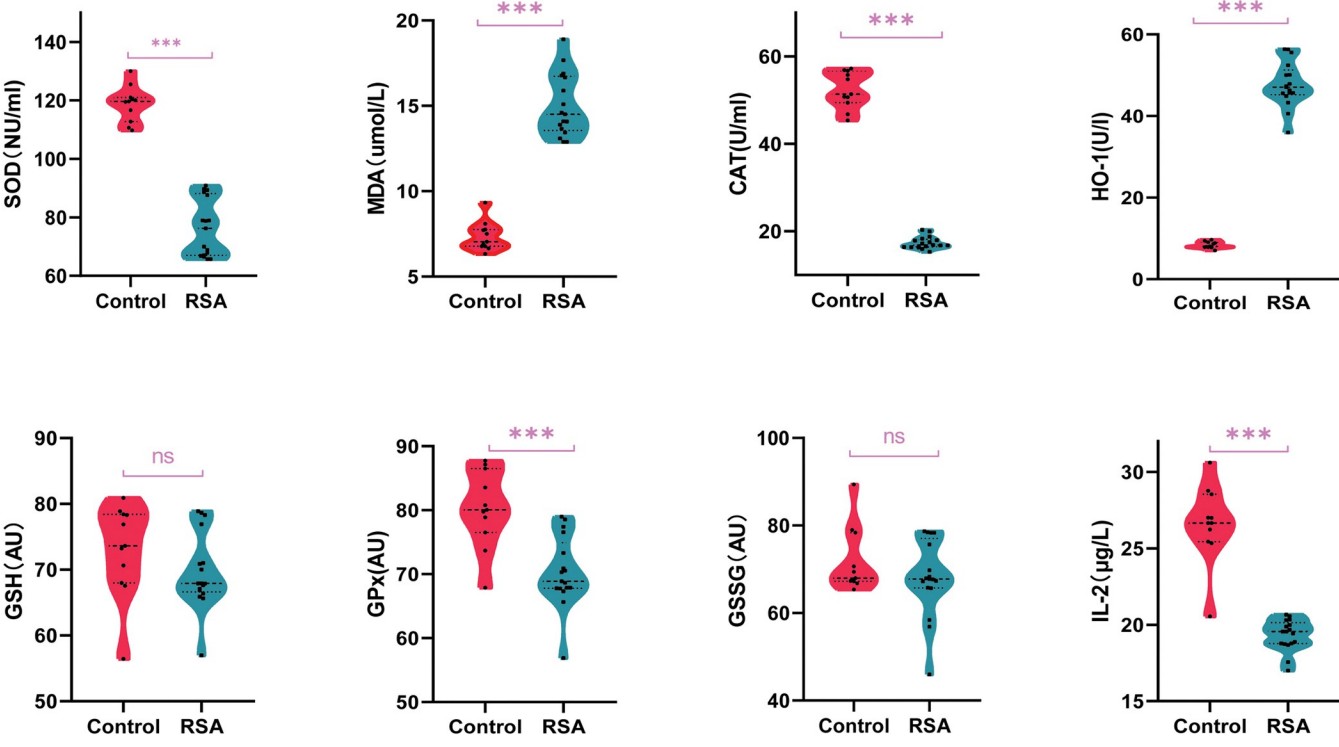

**Fig 6. Results of quantitative analysis of oxidative stress index.**

released from cell membrane phospholipids via phospholipid hydrolysis, and AA-derived compounds such as prostaglandins, leukotrienes (LTs), and thromboxanes, play a crucial role in mediating oxidative stress and regulating hemostasis and irritability [21]. The biosynthesis of leukotrienes, which are derived from AA, involves a two-step process mediated by 5-lipoxygenase. LTs serve important roles in immune regulation, self-defense, and the maintenance of homeostasis in living systems [22]. Additionally, they have been identified as markers of disease outcome [23]. The observed downregulation of LTs in RSA suggests immune system inactivation, this study revealed a significant enrichment of ferroptosis, a newly recognized form of regulated necrotic cell death. Ferroptosis is characterized by lipid oxidation on dysregulation, which is governed by an integrated oxidation and antioxidant system [24]. The metabolism of cysteine and methionine is crucial in the occurrence of ferroptosis. Methionine can be transformed into cystine via desulphurization under oxidative stress, and the synthesis of glutathione (GSH) occurs through the combination of cystine and glutamate, which acts as an antioxidant in the body's response. GPX4, a vital protein for cell survival serves as a core regulator of ferroptosis by degrading lipid peroxides, and inhibiting lipid peroxidation [25]. In the event of decrease in GPX4 expression during this process, the presence of $Fe^{2+}$ iron catalyzes a reaction triggered by phospholipid hydroperoxide, ultimately culminating in cellular demise.

Despite its non-essential status as an amino acid, glutamic acid is prevalent in human blood and serves as a source of carbon and nitrogen for metabolic processes [26, 27]. The mechanisms underlying glutamine metabolism and its regulation in the progression of RSA remain incompletely elucidated. The present study revealed a significant enrichment of glutamate metabolic pathways. The metabolites involved in this pathway, including pyroglutamic acid, glutamic acid and glutamate, exhibited a discernible trend of change in response to the perturbation of RSA. Notably, glutamine assumes a crucial role in energy metabolism by

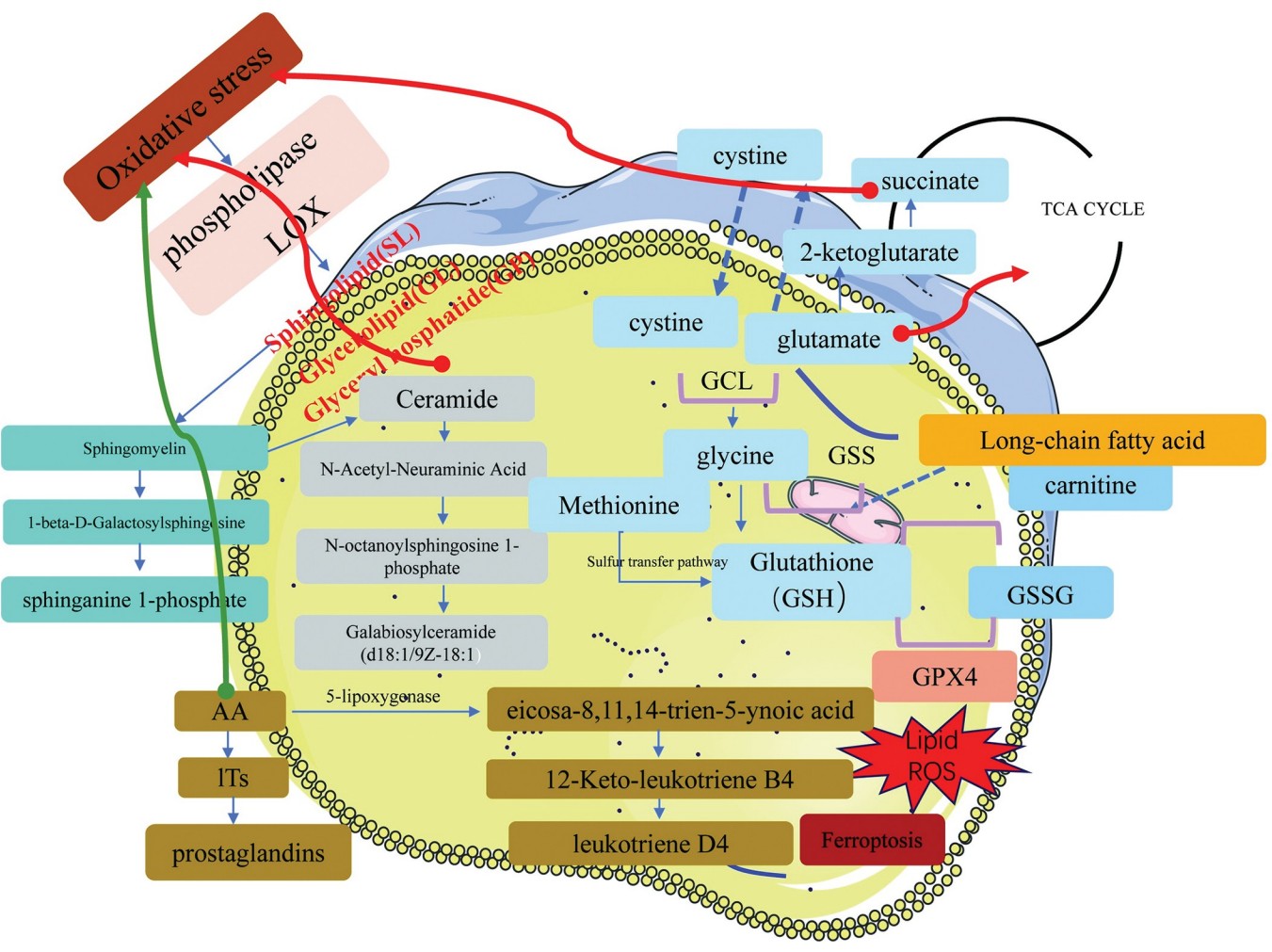

**Fig 7. Major altered pathways in RSA patients.**

replenishing TCA cycle intermediates through a mitochondrial metabolic pathway known as "glutaminolysis"[28]. In the presence of glutaminase (GIs), glutamine undergoes conversion to glutamate, which subsequently yields aketoglutarate (aKG) that enters the TCA cycle and undergoes catabolism to lactate [29]. The onset of RSA elicits body's oxidative stress response, which promptly triggers biochemical reaction and amplifies the energy demand [30]. Alongside TCA energy metabolism, glutamate metabolism plays a direct role in the immune response [31]. Glutamine metabolism provides complementary assistance for macrophage activation and the elicitation of favorable immune responses. Notably, glutaminolysis generates a ketoglutarate (aKG)for alternative (M2) macrophage activation, which stimulates fatty acid oxidation. M1 macrophages, characterized by classical activation, exhibit reduced susceptibility to glutamate metabolism [32]. Upon activation, these macrophages promptly initiate a proinflammatory response to the occurrence of reproductive system-associated events by detecting microbial components, including signaling lipoprotein, and damaged-associated molecular patterns that are released from shed abortive material resulting from sudden symptoms of miscarriage [33]. Additionally, M1 macrophages are closely linked to organism metabolism [34], as they rely on aerobic glycolysis to generate ATP, which leads to increased glucose and glutamine consumption while simultaneously inhibiting metabolism. M1 macrophages

impede the tricarboxylic acid cycle, resulting the buildup of downstream metabolites, including citrate and succinate [35, 36]. Succinate, known for its pro-inflammatory properties, stimulates the production of reactive oxygen species (ROS), thereby exacerbating the onset of disease. The present investigation revealed that alpha-Ketoglutaramate, an upstream metabolite of tartric acid, decreased in the RSA group, indicating the accumulation of tartric acid. Meanwhile, succinate with its pro-inflammatory effects, contributes to the development of disease [37]. Due to the enhancement of glutathione metabolism, an investigation was conducted on the expression of enzyme systems linked to oxidative stress, including GSH, GPx, and GSSG. Heme oxygenase (HO-1), a type of heat shock protein, is among the most readily induced stress-response proteases and is significantly upregulated in cells stimulated by inflammation [38]. The organism can elevate the level of ROS in cells under stimulation, which can stimulate the heightened expression of HO-1, thereby facilitating antioxidant activity and apoptosis regulation through heme catalysis. The alternations in these indices suggested the manifestation of oxidative stress subsequent to abortion and the body's resistance to immune response. IL-2 is capable of preserving the regulatory T cell' activity and contributing to immune regulation. Deviant IL-2 expression signifies the autoimmune system's disproportion [39].

Carnitine, a type of amino acid substance with diverse physiological functions in vivo, assumes a crucial function in the regulation of lipid metabolism [40]. Carnitine and its intermediate binding form possess diverse functional capacities in regulating the oxidative and metabolic status of the female reproductive system. As a carrier of fatty acids, carnitine facilitates the transport of long-chain fatty acids from the mitochondria to the interior of the mitochondria in the form of lipoacylcarnitine, thereby promoting the β-oxidative decomposition of fat [41]. Furthermore, carnitine serves as a scavenger of free radicals, thereby enhancing the antioxidative capabilities that defend against the reduction of oxidative stress [42].

The observed elevation of serum carnitine levels in the RSA group indicated the presence of oxidative stress and the capacity for recovery in response to such stress. The overproduction of ROS can lead to lipid peroxidation and consequent impairment of biofilm integrity, resulting in compromised protein function, nucleic acid and chromosome damage, and ultimately inducing abortion.

Furthermore, T cells holds significant importance as immune cells, primarily tasked with the regulation and mediation of immune response [43]. Within the context of RSA, the activation of T cells can lead to abnormal immune cell activation, inflammatory responses, and embryonic damage through various immunomodulatory pathways. Notably, certain genes play pivotal roles in the activation and regulation of T cells. For instance, LR2 (Toll-like receptor 2) exhibits the ability to modulate immune responses by interacting with T cells within the adaptive immune system [44]. Additionally, CXCL8 (chemokine 8) is implicated in the chemotaxis and activation of T cells [45]. This study exclusively examined the correlation between metabolites and oxidative kinases in RSA. In future investigations, we intend to place greater emphasis on exploring the association between genes and RSA, thereby comprehensively elucidating the systematic occurrence and progression of adverse pregnancy.

Oxidative stress has the potential to induce lipid peroxidation, protein oxidation, and DNA oxidative damage, thereby compromising cellular structure and function. Consequently, comprehending the elucidation of oxidative stress pathways in RSA holds significant importance for therapeutic interventions. Approaches to address oxidative stress encompass enhancing the antioxidant defense system, mitigating the generation of oxidative free radicals, and affording safeguard against oxidative damage. Gene therapy is a potential intervention. By introducing specific genes or gene products, the activity and balance of oxidation pathways can be regulated. For example, by increasing the expression of antioxidant-related genes, antioxidant

capacity can be improved and oxidative stress damage to the embryo can be reduced [46]. The use of antioxidants is widely studied as a potential treatment, such as vitamin C, vitamin E and dopamine [47]. In addition, the degree of oxidative stress can also be reduced by improving lifestyle, such as reducing exposure to oxidative stressors [48].

## Conclusions

The present study 's observation of metabolic changes have the potential to enhance our current limited comprehensive of oxidative stress in RSA. The utilization of metabolomics in conjunction with the assessment of enzyme-related oxidative stress presents a reliable approach for identifying RSA incidence and elucidating its underlying mechanisms. The enrichment pathways analyses have revealed that RSA is intricately linked to lipid metabolism, ferroptosis, carnitine metabolism, and glutathione metabolism. The interrelated metabolic pathways that connect lipid metabolism, energy metabolism, and amino acid metabolism with oxidative stress in the context of abortion are of significant interest to researches. The utilization of metabolomics techniques can offer a more comprehensive understanding of these metabolic processes, thereby facilitating the development of improved diagnostic and therapeutic strategies for RSA. This study had some notably limitations. First, the sample size is small, in particular, in clinical samples. Compared with genes and proteins, metabolites are in the downstream of life activities and are greatly influenced by environment and genetic background factors, and fewer biological repeats may amplify the impact of intra-group differences on the overall differences, thus affecting the final results. In addition, the oxidative stress is a multifaceted and complex process, and defining the best biomarkers for oxidative stress in cells and tissues is challenging. We selected seven oxidative kinases based on enzymes that regulate metabolic disorders, revealing certain limitations in the regulation of oxidative stress in the system.

## Supporting information

**S1 Table. Detailed parameters information of LC-MS.**
(DOCX)

**S2 Table. Coefficient of variation parameter of internal standard.**
(XLSX)

**S3 Table. Detailed clinical information of subjects enrolled in this study.**
(XLSX)

**S4 Table. Identified metabolites of RSA serum in positive and negative mode.MSI Level 1: Metabolites identified by self-built database for standard identification.** Level 2: metabolites identified by public/commercial libraries. Level 3: metabolites identified byMetDNA/AI database. Level 4: metabolites identified by MS fragments.
(XLSX)

**S1 Fig. Analysis of gestational weeks, age and weight in the RSA group compared with the control.** ns: no significance; *: p <0.05; **: p <0.01; ***: p <0.001.
(TIF)

## Acknowledgments

The author thanks engineer ZhiHui Mi and LuLin Song of Inner Inner Mongolia Di An Feng Xin Medical Technology Co., LTD for providing data analysis services.

## Author Contributions

**Conceptualization:** AiNing Wu, RongXin Yu.

**Data curation:** AiNing Wu, YanHui Zhao.

**Formal analysis:** AiNing Wu.

**Investigation:** AiNing Wu.

**Methodology:** RongXin Yu.

**Project administration:** AiNing Wu, Ya Tuo.

**Resources:** AiNing Wu, YanHui Zhao.

**Supervision:** JianXing Zhou.

**Writing – original draft:** AiNing Wu.

**Writing – review & editing:** Ya Tuo.

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
