## [Decision Letter · Decision Letter 0]

27 Sep 2023

PONE-D-23-19156A comprehensive metabolic insight into recurrent spontaneous abortion based on UHPLC-MS/MS for oxidative stress and dysimmunityPLOS ONE

Dear Dr. Tuo,

Thank you for submitting your manuscript to PLOS ONE. After careful consideration, we feel that it has merit but does not fully meet PLOS ONE’s publication criteria as it currently stands. Therefore, we invite you to submit a revised version of the manuscript that addresses the points raised during the review process.

We look forward to receiving your revised manuscript.

Kind regards,

Mohammed Hamdy Farouk, PhD

Academic Editor

PLOS ONE

Journal Requirements:

Reviewers' comments:

Reviewer's Responses to Questions

**Comments to the Author**

1. Is the manuscript technically sound, and do the data support the conclusions?

Reviewer #1: Yes

Reviewer #2: No

2. Has the statistical analysis been performed appropriately and rigorously? 

Reviewer #1: Yes

Reviewer #2: N/A

3. Have the authors made all data underlying the findings in their manuscript fully available?

Reviewer #1: Yes

Reviewer #2: Yes

4. Is the manuscript presented in an intelligible fashion and written in standard English?

Reviewer #1: Yes

Reviewer #2: No

5. Review Comments to the Author

Reviewer #1: It is important to determine gene expression and discuss the RSA at the molecular level.

I suggest joining these findings with the abnormal expression of some specific immunoregulatory genes involved in T-cell activation and differentiation. According to psychophysiological interaction (PPI) analysis, TLR2, CXCL8, IFNG, IL2RA, and ITGAX were among the top five immunoregulatory hub genes of differentially expressed genes in unexplained recurrent spontaneous abortion (URSA). Among the identified differentially expressed genes (DEGs) in URSA, IFNG may play a key role in regulating maternal immune response. AS the key genes and functional pathways identified will provide new insights into the molecular mechanisms involved in RSA pathogenesis and provide potential diagnostic and therapeutic targets.

Reviewer #2: The research titled’’ A comprehensive metabolic insight into recurrent spontaneous abortion based on UHPLC-MS/MS for oxidative stress and dysimmunity ‘’ by Wu et al, investigated the underlying mechanisms contributing to the onset and progression of RSA by identifying potential metabolites as an initial step. This direct clinical study used eleven Serum samples from early pregnancy and 17 from recurrent spontaneous abortion were subjected to liquid chromatography/mass spectrometry (LC-MS) analysis.in addition to measurement of seven oxidative stress products, namely superoxide dismutase , catalase , malonaldehyde , glutathione , glutathione peroxidase , oxidized glutathione , heme oxygenase , was carried out using ELISA.

The current format could be improved by considering the following points:

1-Title: What does the term "dysimmunity" mean?Is it activation or suppression? It is redundant, in my opinion, and needs to be clarified.

2- Abstract:

In conclusion part : Our investigation furnished a more comprehensive analytical framework encompassing metabolites and oxykinase. what do you mean by oxykinase?

3-Introduction:

Both non-volatile targeted metabolites and volatile non-targeted metabolites have been employed on an MS platform to differentiate the RSA and healthy groups (6). The LC platform’s extensive coverage and board dynamic range of metabolites are its primary advantages

Please explain what these terms' abbreviation means.

4-How did you calculate the sample size for this work?

5- What criteria must the patients meet in order to be selected? In terms of age, race, and chronic illnesses like diabetes or other immunological dysfunctions? Metabolomics may be impacted by the factors listed above.

6- Data pretreatment and statistical analysis

To confirm the statistical significance of the differential metabolites, a student’s t-test was conducted between groups, with a p-value less than 0.05 based on OPLS-DA.

Please explain what these terms' abbreviation means.

7- Metabolites identification

AI database ?? Please explain what these terms' abbreviation means.

8- Study participants:

You mentioned that:

‘’The limited sample size precluded the statistical analysis of adverse factors to determine their association with disease’’

Please elaborate. because a greater sample size provides more useful information and lowers odds ratios.

Also, RSA is a prevalent situation that aids in gathering big sample sizes, hence further explanation of the causes of small sample sizes is required.

9- Identification of significantly altered metabolites and pathways

‘’Notably, approximately 200 of these metabolites were associated with drug treatment, likely stemming from prior interventions for pregnant women with RSA. As our investigation did not account for the influence of pharmaceuticals, these metabolites were excluded from our analysis’’

Clarify that please

10- Discussion

- you mentioned that RSA, an autoimmune disease with multifaceted etiology

I disagree that RSA is a disease because it is a disruption of a natural physiological function, possibly caused by an autoimmune condition, during pregnancy.

-The sample size of the cohort (17 patients and 11 healthy controls) was conducted, dynamic changes were captured, resulting in the extraction of dependable e differential information

I think this is a mistake

-Additionally, research has demonstrated that lipid mediators are not only a consequence of oxidative stress but also an inducement for modulating processes (17)

Old information and reference

- The biosynthesis of leukotrienes (LTs), which are derived from AA, involves a two-step process mediated by 5-lipoxygenase. LTs serve important roles in immune regulation, self-defense, and the maintenance of homeostasis in living systems.

-Old data with no reference

-of change in response to the perturbation of RSA. Notably, glutamine assumes a crucial role in energy metabolism by replenishing TCA cycle intermediates through a mitochondrial metabolic pathway known as “glutaminolysis”. In the presence of glutaminase (GIs), glutamine undergoes conversion to glutamate, which subsequently yields aketoglutarate (aKG) that enters the TCA cycle and undergoes catabolism to lactate. The onset of RSA elicits body's oxidative stress response, which promptly triggers biochemical reaction and amplifies the energy demand

where is the references for all of that?

-which stimulates fatty acid oxidation. M1 macrophages, characterized by classical activation, exhibit reduced susceptibility to glutamate metabolism. Upon activation, these macrophages promptly initiate a proinflammatory response to the occurrence of reproductive system associated events by detecting microbial components, including signaling lipoprotein, and damaged-associated molecular patterns that are released from shed abortive material resulting from sudden symptoms of miscarriage.

References??

11- Could you elaborate on the clinical implications of your findings and how it can assist in treating people who have RSA.

12-What is the limitations of your work in frank words

13-Refrences:

-No 2 please check

-Many old references e.g:7,8,9,10 references should be updated

I believe that those comments should be reviewed and corrected before any publication decisions are made.

6. PLOS authors have the option to publish the peer review history of their article (what does this mean?). If published, this will include your full peer review and any attached files.

Reviewer #1: **Yes: **Entsar Rashad Abd-Allah

Reviewer #2: **Yes: **Fatma EL Zharaa Abdelhakam

---

## [Author Response · Author response to Decision Letter 0]

25 Oct 2023

Response to Reviewers

Journal Requirements:

Q: 1. Please ensure that your manuscript meets PLOS ONE's style requirements, including those for file naming. The PLOS ONE style templates can be found at 

A: We have revised the file naming and table naming based on the PLOS ONE templates.

Q:2. We note that the grant information you provided in the ‘Funding Information’ and ‘Financial Disclosure’ sections do not match. When you resubmit, please ensure that you provide the correct grant numbers for the awards you received for your study in the ‘Funding Information’ section.

A: I have corrected the Funding Information.

Reviewer #1:

Q: It is important to determine gene expression and discuss the RSA at the molecular level.

I suggest joining these findings with the abnormal expression of some specific immunoregulatory genes involved in T-cell activation and differentiation. According to psychophysiological interaction (PPI) analysis, TLR2, CXCL8, IFNG, IL2RA, and ITGAX were among the top five immunoregulatory hub genes of differentially expressed genes in unexplained recurrent spontaneous abortion (URSA). Among the identified differentially expressed genes (DEGs) in URSA, IFNG may play a key role in regulating maternal immune response. AS the key genes and functional pathways identified will provide new insights into the molecular mechanisms involved in RSA pathogenesis and provide potential diagnostic and therapeutic targets.

A: Thank you for your suggestions. As suggested by reviewer, we have added the suggested content to the discussion section on page 20.

Reviewer #2:

Q：1. Title：What does the term “dysimmunity” mean? Is it actibation or suppression? It is redundant, in my opinion, and needs to be clarified.

A: Thank you for your suggestions. The title of the manuscript has been revised to “Untargeted metabolomics analysis reveals the metabolic disturbances and exacerbation of oxidative stress in recurrent spontaneous abortion”.

Q:2. Abstract: In conclusion part: our investigation furnished a more comprehensive analytical framework encompassing metabolites and “oxykinase”. What do you mean by oxykinase?

A: The term of “oxykinase” has been revised to “enzymes associated with oxidative stress”.

Q:3. Introduction: Both non-volatile targeted metabolites and volatile non-targeted metabolites have been employed on an MS platform to differentiate the RSA and healthy groups. The LC platform’s extensive coverage and board dynamic range of metabolites are its primary advantages.

Please explain what these terms' abbreviation means.

A: The “LC” and “MS” abbreviation have been revised to “liquid chromatogram” and “mass spectrometer” respectively.

Q:4. How did you calculate the sample size for this work?

A: For metabolomics research, there are the following requirements: the number of experimental samples is larger than that of control samples, and at least 30 human samples are required. However, the number of patients with recurrent fluency is still relatively small in our hospital, which is not specialized in obstetrics. We have done our best to preserve the collection of samples.

Q:5. What criteria must the patients meet in order to be selected? In terms of age, race, and chronic illnesses like diabetes or other immunological dysfunctions? Metabolomics may be impacted by the factors listed above.

A: More detailed information about the exclusion criteria was added in the manuscript on page 5. Statistical analysis has been performed for the age of the enrolled population.

Q:6. Data pretreatment and statistical analysis 

To confirm the statistical significance of the differential metabolites, a student’s t-test was conducted between groups, with a p-value less than 0.05 based on OPLS-DA. Please explain what these terms' abbreviation means.

A: The abbreviation “OPLS-DA” in this sentence has been deleted.

Q:7. Metabolites identification

AI database ?? Please explain what these terms' abbreviation means.

A：The illustration about “AI database” has been added in the manuscript. 

Q:8. Study participants

You mentioned that:

“The limited sample size precluded the statistical analysis of adverse factors to determine their association with disease’’

Please elaborate. because a greater sample size provides more useful information and lowers odds ratios.

Also, RSA is a prevalent situation that aids in gathering big sample sizes, hence further explanation of the causes of small sample sizes is required.

A: The information about the limited sample size was added in the manuscript.

Q:9. Identification of significantly altered metabolites and pathways

“Notably, approximately 200 of these metabolites were associated with drug treatment, likely stemming from prior interventions for pregnant women with RSA. As our investigation did not account for the influence of pharmaceuticals, these metabolites were excluded from our analysis’’

Clarify that please 

A: The detailed information has been clarified in the manuscript on page 12.

Q:10. Discussion

You mentioned that RSA, an autoimmune disease with multifaceted etiology

I disagree that RSA is a disease because it is a disruption of a natural physiological function, possibly caused by an autoimmune condition, during pregnancy. 

A: Thank you for your suggestions. The term disease has been revised to complication

Q:11 The sample size of the cohort (17 patients and 11 healthy controls) was conducted, dynamic changes were captured, resulting in the extraction of dependable e differential information

I think this is a mistake 

A: We agree with the comment and re-wrote the sentences in the revised manuscript as the following “ 17 RSA women patients and 11 healthy pregnant women”. 

Q:12 Additionally, research has demonstrated that lipid mediators are not only a consequence of oxidative stress but also an inducement for modulating processes (17)

Old information and reference

A: The old reference has been replaced with the latest reference in the article.

Q: 13 The biosynthesis of leukotrienes (LTs), which are derived from AA, involves a two-step process mediated by 5-lipoxygenase. LTs serve important roles in immune regulation, self-defense, and the maintenance of homeostasis in living systems.

-Old data with no reference

A: The old reference has been replaced with the latest reference in the article.

Q:14 of change in response to the perturbation of RSA. Notably, glutamine assumes a crucial role in energy metabolism by replenishing TCA cycle intermediates through a mitochondrial metabolic pathway known as “glutaminolysis”. In the presence of glutaminase (GIs), glutamine undergoes conversion to glutamate, which subsequently yields aketoglutarate (aKG) that enters the TCA cycle and undergoes catabolism to lactate. The onset of RSA elicits body's oxidative stress response, which promptly triggers biochemical reaction and amplifies the energy demand

where is the references for all of that?

A: The relevant reference has been inserted into the manuscript.

Q:15 which stimulates fatty acid oxidation. M1 macrophages, characterized by classical activation, exhibit reduced susceptibility to glutamate metabolism. Upon activation, these macrophages promptly initiate a proinflammatory response to the occurrence of reproductive system associated events by detecting microbial components, including signaling lipoprotein, and damaged-associated molecular patterns that are released from shed abortive material resulting from sudden symptoms of miscarriage.

References??

A: The supported reference has been added into the manuscript (reference 32, reference 33).

Q:16 Could you elaborate on the clinical implications of your findings and how it can assist in treating people who have RSA.

A: We are grateful for the suggestion. To be more clearly and in accordance with the reviewer concerns, we have added a more detailed interpretation regarding RSA treating on page 21.

Q:17 What is the limitations of your work in frank words

A: Thank you for your suggestions. As suggested by reviewer, we have added the suggested content to the conclusion section on page 21.

Q:18 Refrences:

-No 2 please check

-Many old references e.g:7,8,9,10 references should be updated

A: The old reference 7, 8, 9, 10 are old reference to the traditional authority detection method, so we did not update the reference, but we updated other older documents in the full paper.

---

## [Decision Letter · Decision Letter 1]

7 Dec 2023

Untargeted metabolomics analysis reveals the metabolic disturbances and exacerbation of oxidative stress in recurrent spontaneous abortion

PONE-D-23-19156R1

Dear Dr. Tuo,

We’re pleased to inform you that your manuscript has been judged scientifically suitable for publication and will be formally accepted for publication once it meets all outstanding technical requirements.

Kind regards,

Mohammed Hamdy Farouk, PhD

Academic Editor

PLOS ONE

Additional Editor Comments (optional):

Reviewers' comments:

Reviewer's Responses to Questions

**Comments to the Author**

1. If the authors have adequately addressed your comments raised in a previous round of review and you feel that this manuscript is now acceptable for publication, you may indicate that here to bypass the “Comments to the Author” section, enter your conflict of interest statement in the “Confidential to Editor” section, and submit your "Accept" recommendation.

Reviewer #1: All comments have been addressed

Reviewer #2: All comments have been addressed

2. Is the manuscript technically sound, and do the data support the conclusions?

Reviewer #1: Yes

Reviewer #2: Yes

3. Has the statistical analysis been performed appropriately and rigorously? 

Reviewer #1: Yes

Reviewer #2: Yes

4. Have the authors made all data underlying the findings in their manuscript fully available?

Reviewer #1: Yes

Reviewer #2: Yes

5. Is the manuscript presented in an intelligible fashion and written in standard English?

Reviewer #1: Yes

Reviewer #2: Yes

6. Review Comments to the Author

Reviewer #1: the article can be accepted , but i hope to complete this study involving immunoregulatory genetic parameters.

Reviewer #2: The authors develop a unique theoretical framework, and I believe that they should highlight their originality much more.

7. PLOS authors have the option to publish the peer review history of their article (what does this mean?). If published, this will include your full peer review and any attached files.

Reviewer #1: **Yes: **Entsar Rashad Abd-Allah

Reviewer #2: **Yes: **Fatma EL Zahraa Abd EL Hakam

---

## [Editor Report · Acceptance letter]

13 Dec 2023

PONE-D-23-19156R1 

PLOS ONE

Dear Dr. Tuo, 

I'm pleased to inform you that your manuscript has been deemed suitable for publication in PLOS ONE. Congratulations! Your manuscript is now being handed over to our production team.

Kind regards, 

on behalf of

Prof. Mohammed Hamdy Farouk 

Academic Editor

PLOS ONE